# FEATURE-BASED METRICS FOR EXPLORING THE LATENT SPACE OF GENERATIVE MODELS

**Samuli Laine**
NVIDIA
slaine@nvidia.com

## ABSTRACT

Several recent papers have treated the latent space of deep generative models, e.g., GANs or VAEs, as Riemannian manifolds. The argument is that operations such as interpolation are better done along geodesics that minimize path length not in the latent space but in the output space of the generator. However, this implicitly assumes that some simple metric such as $L_2$ is meaningful in the output space, even though it is well known that for, e.g., semantic comparison of images it is woefully inadequate. In this work, we consider imposing an arbitrary metric on the generator's output space and show both theoretically and experimentally that a feature-based metric can produce much more sensible interpolations than the usual $L_2$ metric. This observation leads to the conclusion that analysis of latent space geometry would benefit from using a suitable, explicitly defined metric.

## 1 INTRODUCTION

Let us assume a deterministic generator function $g : \mathcal{Z} \to \mathcal{X}$, for example the generator of a trained GAN (Goodfellow et al., 2014). The latent space $\mathcal{Z} = \mathbb{R}^d$ may be constrained, e.g., by requiring $\mathbf{z}$ to have a constant norm (Karras et al., 2018), or to be drawn from a specific probability distribution (Kingma & Welling, 2014). The output space $\mathcal{X} \subseteq \mathbb{R}^D$ typically has much higher dimension than the latent space. Suppose we are given two latent vectors $\mathbf{z}_0$ and $\mathbf{z}_1$ and wish to construct an interpolation function $\gamma(t)$, $0 \le t \le 1$, such that $\gamma(0) = \mathbf{z}_0$, $\gamma(1) = \mathbf{z}_1$, and $g(\gamma(t))$ is somehow smooth. A trivial choice is to set $\gamma(t) = \mathbf{z}_0 + t(\mathbf{z}_1 - \mathbf{z}_0)$, but this is completely oblivious to the operation of $g$. With typical generators, we cannot say much more about $g$ than that it is continuous, so the interpolation result may be visually disappointing.

The trivial interpolator above minimizes the path energy $\int ||\partial \gamma(t)/\partial t||^2 \, \mathrm{d}t$ in $\mathcal{Z}$. Shao et al. (2017) suggest that a more principled quantity to minimize is the energy of the path in $\mathcal{X}$, i.e., $\int ||\partial g(\gamma(t))/\partial t||^2 \, \mathrm{d}t$. Alternatively, dropping the square in the formula, we obtain the path length that can be similarly minimized (Chen et al., 2017; Arvanitidis et al., 2018). However, the path length has a trivial minimum if we attempt to approximate the path with a discrete curve with $n$ vertices and approximate $\partial g(\cdot)/\partial t$ by differences — it is always optimal to bunch all intermediate path vertices at the endpoints, thereby reducing the curve to a line segment in $\mathcal{X}$ that ignores the manifold spanned by $g$. Placing a vertex anywhere else, forming a series of two line segments, can only increase the path length due to triangle inequality. As such, we prefer to work with path energy instead of path length.

It is worth asking what minimizing the above norm $|| \cdot ||^2$ over the curve achieves for a generator that outputs images. The "optimal" interpolation between two images, in terms of path energy in $\mathcal{X}$, would be a linear cross-fade between them. Hence, the result of the minimization is an image sequence that is as close to a cross-fade that the generator can produce! If we were to optimize path length instead, the optimum would be similar except that the progression of the fade would not need to be linear. The cross-fade tendency is visible in, e.g., Figure 7 (bottom) of Arvanitidis et al. (2018).

Clearly, any simple norm over differences (or time derivatives) of images is inadequate for defining a "meaningful" interpolation. Therefore, we propose that instead of minimizing the path energy in either $\mathcal{Z}$ or $\mathcal{X}$, we minimize the energy of an explicit metric that can be chosen to suit the task at hand. In the continuous case, we would thus minimize $\int m(g(\gamma(t)), \partial g(\gamma(t))/\partial t)^2 \, \mathrm{d}t$ where $m$ assigns a non-negative cost to the gradient of the generator at a given point. For the discrete case, we

can similarly minimize $\sum_{i=1}^{n-1} m(g(\mathbf{z}_i), g(\mathbf{z}_{i+1}))^2$, where $\mathbf{z}_{1 \ldots n}$ are the vertices of a discrete path in $\mathcal{Z}$, and $m : \mathcal{X}, \mathcal{X} \to \mathbb{R}_{\geq 0}$ takes two images and measures their difference. As long as $m$ and $g$ are differentiable, we can minimize this expression end-to-end by treating all but the first and last $\mathbf{z}_i$ as optimizable parameters.

## 2 EXPERIMENTS

In all of our tests, our $g(\cdot)$ is a pre-trained generator of the progressive GAN of Karras et al. (2018) trained using the CelebA-HQ dataset. The output of the generator is a $1024 \times 1024$-pixel RGB image. Our feature-based metric is based on the activations of a VGG-19 network (Simonyan & Zisserman, 2014) pre-trained on ILSVRC-2014 (Russakovsky et al., 2015). The same network has been used for similar purposes in, e.g., texture synthesis (Gatys et al., 2016), image manipulation (Upchurch et al., 2017), image synthesis from semantic segmentation (Chen & Koltun, 2017), and estimation of perceptual similarity of images (Zhang et al., 2018).

Because the VGG-19 we use was originally trained for $224 \times 224$ images, we downsample the generated images to $256 \times 256$ resolution before presenting them to the network. The generator's output range $[-1, 1]$ is also mapped to the input range of the VGG-19 network. Following Chen & Koltun (2017), we extract the activations of layers 'conv1_2', 'conv2_2', 'conv3_2', 'conv4_2', and 'conv5_2' in the VGG-19 network. Denoting the output tensor of layer conv$j$_2 for input image $\mathbf{x}$ as $V_j(\mathbf{x})$, we compute the difference between two images as:

$$m_{VGG}(\mathbf{x}_1, \mathbf{x}_2) = \sum_{j=1}^{5} \frac{1}{N_j} ||V_j(\mathbf{x}_1) - V_j(\mathbf{x}_2)||^2,$$

where $N_j$ denotes the number of scalars in the layer output. As comparison methods we use pixel-space MSE and linear interpolation in $\mathcal{Z}$. The pixel-space MSE is also computed in $256 \times 256$ resolution so that it cannot pick up microstructure that is invisible to the feature-based metric. Because our generator is trained on normalized $\mathbf{z}$, we restrict our path vertices on the appropriate hypersphere during optimization. The same holds for the linear interpolation comparison. All paths we optimize have 32 segments, i.e., 33 vertices. For optimization we use Adam (Kingma & Ba, 2015) with learning rate $\lambda = 0.01$ and otherwise default parameters.

**Progressive path subdivision.** To accelerate the process, we subdivide the path progressively during optimization. We start with two segments, i.e., one free vertex placed at the midpoint in $\mathcal{Z}$, and optimize it for 50 iterations. Then we subdivide both segments similarly and optimize the three free vertices for 50 epochs, etc. To reach 32 segments, we thus run the optimization for a total of 250 iterations. Constructing a path this way takes approximately 3 minutes on a single GPU.

**Brightness and contrast equalization.** If a generator is versatile enough, it is likely able to produce variations of the same image under different lighting conditions that lead to varying levels of brightness and contrast. This is true for our generator as well, which reveals some interesting failure modes for the metrics. Figure 1 shows that minimizing MSE in $\mathcal{X}$ (row 2) can indeed exhibit the predicted cross-fade phenomenon, leading to low image contrast at the middle of path. However, $m_{VGG}$ (row 3) is similarly suspect to darkening the image towards the middle rather than keeping the features consistent, as this apparently minimizes the differences in VGG-19 network activations.

To remedy this, we can equalize the brightness and contrast of the images — specifically, offset and scale the image to zero mean and unit variance — before evaluating the metrics. See Figure 1, rows 4–5, for results (metrics with image equalization are denoted with a tilde). We observe that the contrast-flattening tendency of $\mathcal{X}$ MSE is somewhat lessened, and that our feature-based metric does not "cheat" anymore by passing through dark images. Note that the images in the figure have not been equalized but are shown exactly as they were output by $g$; the equalization only affects the inputs to the metrics, and thus the energy landscape of the path optimization process.

**Conclusion.** Figure 2 shows further interpolation results. We can see that the $\widetilde{m}_{VGG}$ metric achieves the most natural-looking interpolations in these cases, whereas both comparison methods are prone to hallucinating features that are not present at either endpoint. It seems clear, both theoretically and experimentally, that we cannot rely on a simple norm in the generator output space to impose a meaningful structure to the latent space. Therefore, our view is that further analysis of the geometry of the latent space would benefit from using a suitable, explicitly defined metric.

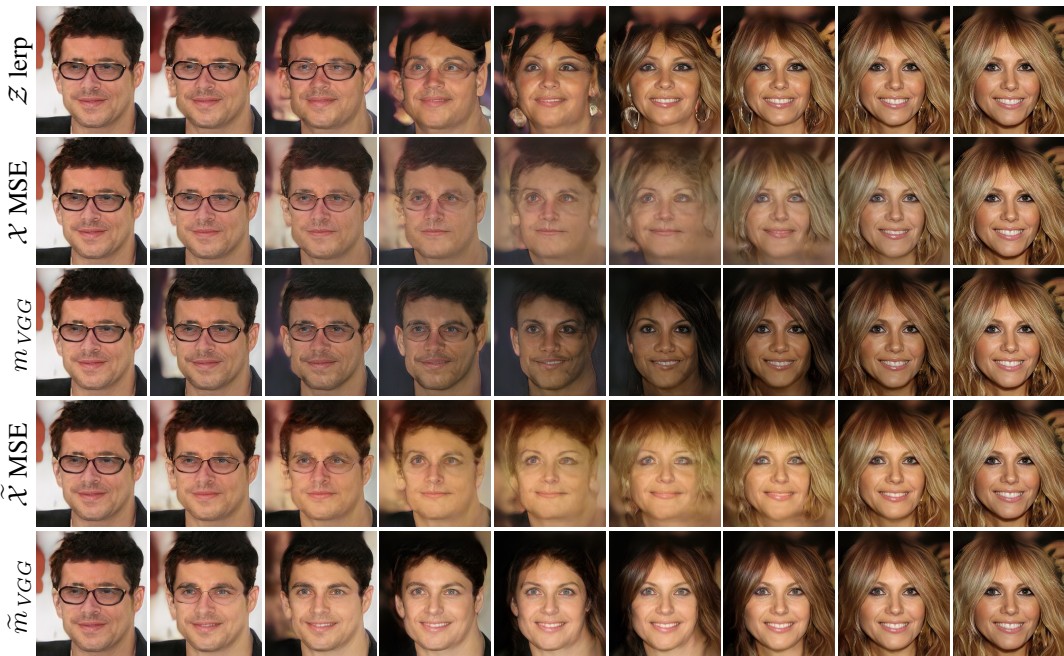

Figure 1: Comparison of five different interpolation schemes. Row 1 shows the baseline naïve interpolation in $\mathcal{Z}$, and row 2 corresponds to previous work. Rows 3–5 correspond to methods proposed in this paper, see text for details.

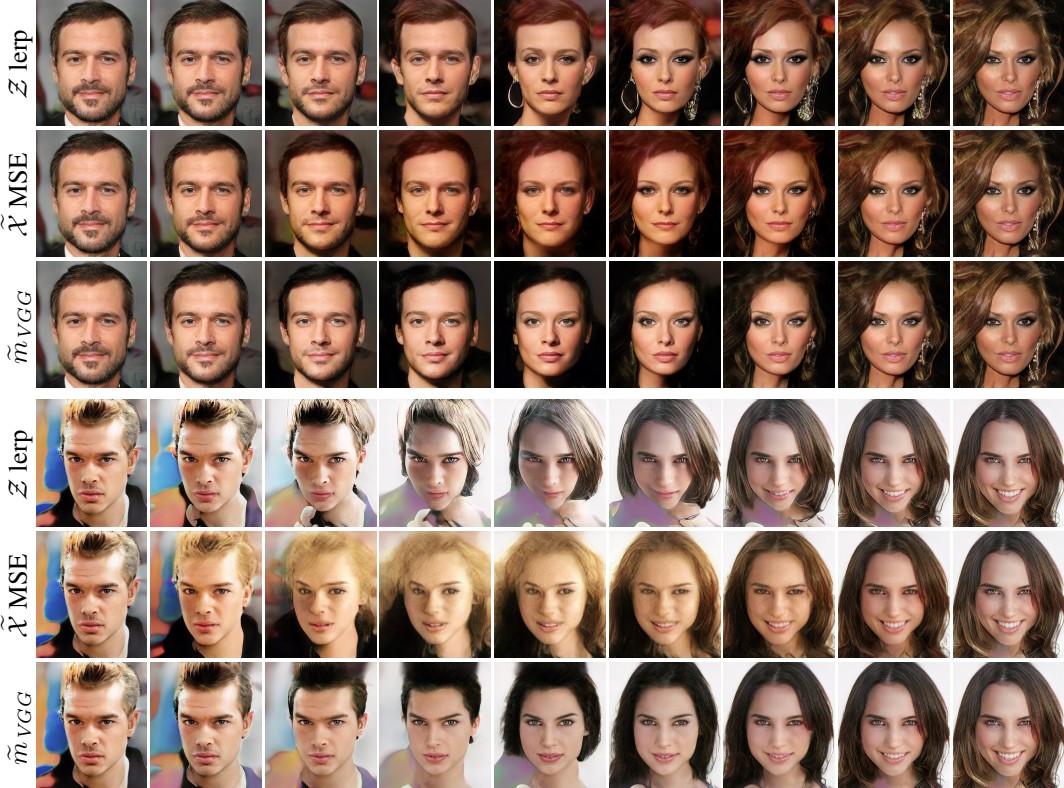

Figure 2: Further interpolation results. Both naïve $\mathcal{Z}$ interpolation and $\widetilde{\mathcal{X}}$ MSE based energy minimization are prone to passing through features that are not present at either endpoint, while $\widetilde{m}_{VGG}$ tends to avoid this phenomenon.

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
