# OpenReview forum: "Feature-Based Metrics for Exploring the Latent Space of Generative Models"
_ICLR.cc/2018/Workshop — Accept_

### Official Review · AnonReviewer2 · 2018-02-24
**Sensible work on the importance of using the right metric in generative models**

**Rating:** 7
**Confidence:** 5

**Review:**

The paper builds on recent work that views generative models as learning a Riemannian manifold. This recent work then pulls the (Euclidean) metric of the output space into the latent space, and performs interpolation along geodesics under such a metric. The present paper argues that pulling the Euclidean metric into the latent space is not suitable for perceptual tasks as this metric is unrelated to perception. It is then proposed to instead pull back a feature-based metric, where the features are extracted from a VGG-19 network. A few examples indicate that this gives nicer interpolations.

I'm generally positive towards this work, but have a few comments:

*) The present paper rely on a deterministic generator, so it does not describe basic models such as the VAE even if this is hinted at in the introduction. Arvanitidis et al. has shown that using stochastic generators gives significantly more meaningful metrics, than when using deterministic generators. It would be good to rely on stochastic generators as the metric of deterministic generators appear to be arbitrarily bad, which renders the comparative study unfair. That being said, I think the author's conclusion is perfectly sensible.

*) In the introduction it is stated that it is shown theoretically that choosing another metric to pull back is better. I did not find such a theoretical statement in the paper, so I'd suggest to tone down the claims.

---

### Official Review · AnonReviewer3 · 2018-03-05
**Interesting, small but good contribution to latent space geometry**

**Rating:** 7
**Confidence:** 4

**Review:**

The paper builds on recent literature that explore the Riemannian geometry of latent space representations for e.g. improved interpolation. While previous work use the pullback of the data space Euclidean metric, the present paper proposes that pullbacks of other metrics to the latent space can give better results. This is tested using a feature based metric by visual comparison of interpolation between test images.

The idea of using other metrics than pullback of Euclidean metrics is well-founded and novel. The paper is well-written and clear. While the contribution is incremental, I believe progressing a step further than the basic geometry explored in previous latent space geometry papers is important and needed.

I my view, the the paper can be accepted because of a novel and well-founded contribution that, although incremental, adds to the geometric exploration of latent space models. The experimental validation is limited which is the major con of the paper.

---

### Official Review · AnonReviewer1 · 2018-03-11
**Interesting idea but limited evaluation**

**Rating:** 7
**Confidence:** 4

**Review:**

PAPER SUMMARY
The paper discusses methods for interpolating between images in the latent space of GAN generators. The straightforward approach is linear interpolation in the latent space; some recent work has suggested minimizing path energy in image space. This paper instead considers finding paths which minimize path energy in VGG feature space using gradient descent on the path itself. The paper shows qualitative examples where this gives nicer results than other sorts of latent-space interpolation.

Pros
(+) The idea of using VGG features (or other arbitrary differentiable image similarity metric) for latent-space exploration is a nice idea, and novel to the best of my knowledge
(+) The interpolations using the proposed method do seem qualitatively different from the results using other interpolation methods

Cons
(+) The only results are a handful of qualitative results on CelebA-HQ; I would have also liked to see results on other datasets, such as MNIST or CIFAR which are frequently used for GANs.
(+) No quantitative evaluation. Latent-space interpolation is a hard thing to quantify, but one idea would be to run a user study to quantify which method produces interpolations that are preferred by users over a large set of images.

Questions
- For the generator used in this paper, were latent vectors constrained to unit norm? If so I also would have liked to see a comparison with interpolation along the surface of a hypersphere in latent space.

Overall the paper presents an interesting and novel idea, but backs it up with only a few qualitative examples on a single dataset; this makes it hard to assess the general applicability of the method. Nevertheless the method is likely to be of interest to the community, and seems like a good fit for the workshop track.

---

### Decision · Program_Chairs · 2018-03-20
**ICLR 2018 Workshop Acceptance Decision**

**Decision:**

Accept

**Comment:**

Congratulations, your paper was accepted to the ICLR workshop.

---

> ### Public Comment · ~Samuli_Laine1 · 2018-03-22
> **Thank you**
>
> I'd like to thank all reviewers for the constructive comments. I will revise the paper according to the suggestions. If at some point this were to be fleshed out into a more complete publication, I agree that more datasets and generator types would be needed, but that will have to wait until after the conference.
>
> To answer the question of reviewer #1, the generator internally normalizes the latent vector, and thus all generated images are from the distribution it was trained with. With this normalization, the only difference between Euclidean and hypersphere interpolation would be the acceleration profile along the path.